# An Auto-Encoder Strategy for Adaptive Image Segmentation

**Evan M. Yu**[1]                                                                    EMY24@CORNELL.EDU

**Juan Eugenio Iglesias** [2,3,4]                                               E.IGLESIAS@UCL.AC.UK

**Adrian V. Dalca** [2,3]                                                             ADALCA@MIT.EDU

**Mert R. Sabuncu** [1,5]                                                         MSABUNCU@CORNELL.EDU

[1] *Nancy E. and Peter C. Meinig School of Biomedical Engineering, Cornell University*

[2] *Martinos Center for Biomedical Imaging, Massachusetts General Hospital, Harvard Medical School*

[3] *Computer Science and Artificial Intelligence Laboratory (CSAIL), MIT*

[4] *Centre for Medical Image Computing, University College London*

[5] *School of Electrical and Computer Engineering, Cornell University*

## Abstract

Deep neural networks are powerful tools for biomedical image segmentation. These models are often trained with heavy supervision, relying on pairs of images and corresponding voxel-level labels. However, obtaining segmentations of anatomical regions on a large number of cases can be prohibitively expensive. Thus there is a strong need for deep learning-based segmentation tools that do not require heavy supervision and can continuously adapt. In this paper, we propose a novel perspective of segmentation as a discrete representation learning problem, and present a variational autoencoder segmentation strategy that is flexible and adaptive. Our method, called Segmentation Auto-Encoder (SAE), leverages all available unlabeled scans and merely requires a segmentation prior, which can be *a single unpaired* segmentation image. In experiments, we apply SAE to brain MRI scans. Our results show that SAE can produce good quality segmentations, particularly when the prior is good. We demonstrate that a Markov Random Field prior can yield significantly better results than a spatially independent prior. Our code is freely available at https://github.com/evanmy/sae.

**Keywords:** Image Segmentation, Variational Auto-encoder

## 1. Introduction

Quantitative biomedical image analysis often builds on a segmentation of the anatomy into regions of interest (ROIs). Recently, deep learning techniques have been increasingly used in a range of segmentation applications (Akkus et al., 2017; Litjens et al., 2017; Ronneberger et al., 2015; Kamnitsas et al., 2017). These methods often rely on a large number of *paired* scans and segmentations (voxel-level labels) to train a neural network. Training labels are either generated by human experts, which can be costly and/or hard to scale, or automatic software (Dolz et al., 2018), which can constrain performance. Furthermore, supervised techniques typically yield tools that are sensitive to changes in image characteristics, for instance, due to a modification of the imaging protocol (Jog and Fischl, 2018). This is a significant obstacle for the widespread clinical adoption of these technologies.

One approach to improve robustness and performance is to relax the dependency on paired training data and simply use unpaired examples of segmentations, sometimes called "atlases." Building on unpaired atlases, a segmentation model can then be trained continuously on new sets of unlabeled images (Dalca et al., 2018b, 2019; Joyce et al., 2018). For example, recently Dalca *et al.* (Dalca et al., 2018b) proposed an approach where an autoencoder is pre-trained on *thousands* of unpaired atlases. For a new set of unlabeled images, the encoder is then re-trained via an unsupervised strategy. Another widely-used approach to improve generalizability is data augmentation on labeled training data (Zhao et al., 2019; Chaitanya et al., 2019). For example, Zhao et al. 2019 demonstrated an adaptive approach that learns an augmentation model on a dataset of unlabeled images. This model was then applied to augment a single paired atlas to perform one-shot segmentation within a supervised learning framework. Another popular approach is to use registration to propagate atlas labels to a test image (Sabuncu et al., 2010; Lee et al., 2019).

In this paper, we present a novel perspective for *minimally supervised* image segmentation. Instead of viewing segmentation from the lens of supervised learning or inverse inference, we regard it as a discrete representation learning problem, which we solve with a variational autoencoder (VAE) like strategy (Kingma and Welling, 2013). We call our framework Segmentation Auto-encoder, or SAE. As we demonstrate below, SAE is flexible and can leverage *all* available data, including unpaired atlases and unlabeled images. We show that we can train a good segmentation model using SAE with as little as a *single unpaired* atlas. In conventional representation learning, e.g., VAE (Kingma and Welling, 2013), an encoder maps an input to a continuous latent representation, which often lacks interpretability. In contrast, in SAE, the encoder computes a discrete representation that is a segmentation image, which is guided by an atlas prior. Finally, we employ the Gumbel-softmax relaxation (Jang et al., 2016) to train the SAE network. The Gumbel-softmax approximates the non-differentiable argmax (tresholding) operation with a softmax in order to make the function differentiable. It provides us with a simple and efficient way to perform the reparameterization trick for a categorical distribution, allowing the network to be trained via back-propagation. In our experiments, we demonstrate that SAE produces high quality segmentation maps, even with a single unpaired atlas. We also quantify the boost in performance as we exploit richer prior models. For example, a Markov Random Field model yields significantly better results than a spatially independent prior.

## 2. Method

We consider a dataset of $N$ observed images (e.g. MRI scans) $\{\boldsymbol{x}^{(i)}\}_{i=1}^N$, which we model as independent samples from the same distribution. Let $\boldsymbol{s}$ denote the (latent) segmentation, where each voxel is assigned a unique discrete anatomical label. Using Bayes' rule:

$$\log p(\boldsymbol{x}^{(i)}) = \log \sum_{\boldsymbol{s}} p(\boldsymbol{x}^{(i)}|\boldsymbol{s})p(\boldsymbol{s}), \tag{1}$$

where $p(\boldsymbol{s})$ denotes a prior distribution on the segmentation, $p(\boldsymbol{x}^{(i)}|\boldsymbol{s})$ is the posterior probability of the observed image conditioned on the latent segmentation, often called the image likelihood, and the sum is over all possible values of $\boldsymbol{s}$. We assume the prior $p(\boldsymbol{s})$ is provided and "learning" involves finding the parameters that describe the image likelihood $p(\boldsymbol{x}^{(i)}|\boldsymbol{s})$.

Since Eq. (1) is computationally intractable for most practical scenarios, we follow the classical variational strategy and maximize the evidence lower bound objective (ELBO):

$$\log p(\boldsymbol{x}^{(i)}) \geq -\text{KL}(q(\boldsymbol{s}|\boldsymbol{x}^{(i)})||p(\boldsymbol{s})) + \mathbb{E}_{\boldsymbol{s}\sim q(\boldsymbol{s}|\boldsymbol{x}^{(i)})} \log p(\boldsymbol{x}^{(i)}|\boldsymbol{s}), \tag{2}$$

where $\text{KL}(\cdot||\cdot)$ denotes the KL-divergence and $q(\boldsymbol{s}|\boldsymbol{x}^{(i)})$ is an efficient-to-manipulate distribution that approximates the true posterior $p(\boldsymbol{s}|\boldsymbol{x}^{(i)})$.

Following the VAE (Kingma and Welling, 2013) framework, we use two neural networks to compute the approximate posterior $q(\cdot|\cdot)$ and the image likelihood $p(\cdot|\cdot)$. A so-called encoder network computes the approximate posterior $q_\phi(\boldsymbol{s}|\boldsymbol{x})$, where $\phi$ denotes the parameters of the encoder. The image likelihood $p_\theta(\boldsymbol{x}|\boldsymbol{s})$ is computed by the a decoder network, parameterized by $\theta$. In our formulation, the encoder can be viewed as a segmentation network. The decoder corresponds to a generative or "reconstruction" model that describes the process of creating an observed image from an underlying segmentation.

A natural choice for the approximate posterior is a voxel-wise independent model:

$$q_\phi(\boldsymbol{s}|\boldsymbol{x}^{(i)}) = \prod_{j=1}^{V} \text{Cat}(s_j|\boldsymbol{x}^{(i)}, \phi), \tag{3}$$

where $\text{Cat}(s_j|\boldsymbol{x}^{(i)}, \phi)$ is a categorical distribution computed as the soft-max output of the encoder network at the $j^{\text{th}}$ voxel evaluated for label $s_j$. Assuming an additive Gaussian noise likelihood model:

$$p_\theta(\boldsymbol{x}|\boldsymbol{s}) = \prod_{j=1}^{V} \mathcal{N}(\boldsymbol{x}; \hat{\boldsymbol{x}}_j(\boldsymbol{s}; \theta), \sigma^2), \tag{4}$$

where $\hat{\boldsymbol{x}}(\boldsymbol{s}; \theta)$ is a "reconstruction" image computed by the decoder network, sub-script $j$ is the voxel index, and $\mathcal{N}(\cdot; \mu, \sigma^2)$ denotes a Gaussian with mean $\mu$ and variance $\sigma^2$.

Putting together Eq. (2) and (4) and relying on Monte Carlo sampling to approximate the expectation, we obtain the following loss function to be minimized over $\theta$ and $\phi$:

$$\mathcal{L} = \sum_{i=1}^{N} \text{KL}(q_\phi(\boldsymbol{s}|\boldsymbol{x}^{(i)})||p(\boldsymbol{s})) + \frac{V}{2}\log\sigma^2 + \frac{1}{2\sigma^2 K}\sum_{k=1}^{K}||\boldsymbol{x}^{(i)} - \hat{\boldsymbol{x}}(\boldsymbol{s}_{ik}; \theta)||_2^2, \tag{5}$$

where $\boldsymbol{s}_{ik}$ is an independent sample segmentation image drawn from $q_\phi(\boldsymbol{s}|\boldsymbol{x}^{(i)})$. Following the convention in the field, in practice we set $K = 1$, which yields an unbiased yet noisy estimate of the loss and its gradient. Eq. 5 does not explicitly require paired images and segmentations $\{\boldsymbol{x}^{(i)}, \boldsymbol{s}^{(i)}\}$. Instead, it merely needs a prior $p(\boldsymbol{s})$. There are many ways to define a prior, but in our experiments we use a classical construction: a probabilistic atlas that describes the probability of labels at each location, which can be coupled with a Markov random field component that encourages certain topological arrangements.

## 2.1. Spatial Prior

The first prior we consider is a probabilistic atlas that assigns an independent label probability vector at each voxel, $p_j$. We call this a spatial prior:

$$p_{spatial}(\boldsymbol{s}) = \prod_{j=1}^{V} p_j(s_j). \tag{6}$$

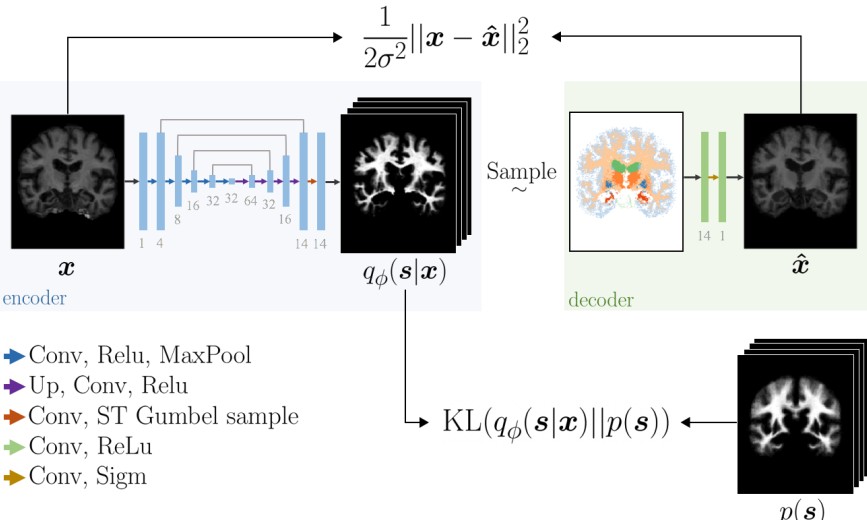

Figure 1: **Proposed architecture**. The encoder (blue) is a U-Net and decoder (green) is a simple CNN. (Conv) 3x3x3 convolution (Relu) rectified linear unit (Maxpool) 2x downsample (Up) 2x upsample (ST Gumbel) straight through Gumbel softmax (Sigm) sigmoid. The number of channels are displayed below each layer.

There are many ways to construct this type of prior. For example, we can aggregate segmentations of different subjects and compute the frequency of anatomical labels at each voxel. If instead we only have a single segmentation image, we can apply a spatial blur to this segmentation in order to account for inter-subject variation. With the spatial prior, the first term in Eq. (5) reduces to:

$$\text{KL}(q_\phi(\boldsymbol{s}|\boldsymbol{x}^{(i)})||p_{spatial}(\boldsymbol{s})) = \sum_{j=1}^{V} H(\text{Cat}(s_j|\boldsymbol{x}^{(i)}), p_j(s_j)) - H(\text{Cat}(s_j|\boldsymbol{x}^{(i)})) \tag{7}$$

where the first term denotes cross-entropy and second term is marginal entropy.

## 2.2. Markov Random Field Prior

The spatial prior can be modified using a Markov Random Field (MRF) to capture neighborhood relationships in a segmentation image. Following (Zhang et al., 2001; Fischl et al., 2002), we define the MRF prior as:

$$p_{MRF}(\mathbf{s}) = \frac{1}{Z} \exp\left[ \sum_{j=0}^{V} V_j(s_j) + \sum_{j=0}^{V} \sum_{k \in N_j} V(s_k, s_j) \right] \tag{8}$$

where $N_j$ is the $3 \times 3 \times 3$-neighborhood around voxel $j$, $V_j(\cdot)$ is the unitary potential at voxel $j$, $V(\cdot, \cdot)$ is the pairwise clique potential, and $Z$ is a normalization constant. Similar to (Fischl et al., 2002), we define these potential functions based on a provided probabilistic

atlas. Specifically, $V_j$ is the voxelwise log frequency of each label: $\log p_j$; and $V(\cdot, \cdot)$ is the log normalized counts of label co-occurrences in neighboring voxels. E.g., $V(l_1, l_2)$ is computed as the logarithm of the count of neighboring voxel pairs with labels $l_1$ and $l_2$ divided by the count of voxels with label $l_2$. If the pairwise potential is set to zero, the MRF prior reduces to the spatial prior. With the MRF prior, the first term in Eq. (5) becomes:

$$\text{KL}(q_\phi(\boldsymbol{s}|\boldsymbol{x}^{(i)})||p_{MRF}(\boldsymbol{s})) = \text{KL}(q_\phi(\boldsymbol{s}|\boldsymbol{x}^{(i)})||p_{spatial}(\boldsymbol{s})) + \mathcal{L}_{MRF} + \text{const.}, \qquad (9)$$

where the first term is from Eq. (7), and the second term can be expressed as:

$$\mathcal{L}_{MRF} = -\sum_{j=0}^{V}\left(\sum_{l_j=0}^{L-1}q_j(l_j|\mathbf{x}^{(i)})\sum_{l_k=0}^{L-1}\sum_{k\in\mathcal{N}_j}q_y(l_k|\mathbf{x}^{(i)})V(s_k=l_k, s_j=l_j)\right). \qquad (10)$$

The MRF loss term quantifies the dissimilarity between the label topology of the prior and the approximate posterior $q(\cdot|\cdot)$.

### 2.3. Implementation Details

Our SAE architecture is shown in Fig. 1. The encoder is a 3D U-Net (Ronneberger et al., 2015) and the decoder is a simple fully convolutional network. Training involves optimizing Eq. (5) with back-propagation. To implement the sampling layer, we employed the straight-through Gumbel-softmax relaxation scheme (Jang et al., 2016; Maddison et al., 2016), with the recommended setting for the temperature $\tau$ to 2/3. We estimated $\sigma^2$ by using the the global mean square error (MSE) between the reconstructed scan $\hat{\boldsymbol{x}}$ and the input scan $\boldsymbol{x}^{(i)}$. To initialize $\sigma^2$, we set the weight on the the reconstruction loss to be zero for the first 16 subjects (effectively setting $\sigma^2$ to infinity) so that the segmentation (encoder) network was trained only based on the prior. In subsequent batches, $\sigma^2$ was updated as the average MSE over the latest 16 subjects and rounded to the nearest power of 10 in order to reduce fluctuation. Our complete model is trained end-to-end with the ADAM optimizer (Kingma and Ba, 2014), with a learning rate of $10^{-4}$ and default parameter for its first and second moments. At test time, segmentation involves a computationally efficient single forward pass through the encoder and we output the `argmax` label at each voxel. Our code in PyThorch is available at https://github.com/evanmy/sae.

## 3. Experiments

### 3.1. Dataset

We evaluated SAE on T1-weighted 3D brain MRI scans, which we preprocessed with FreeSurfer, including skull stripping, bias-field correction, intensity normalization, affine registration to Talairach space, and resampling to 1 $mm^3$ isotropic resolution (Fischl, 2012). We focused on 12 brain regions (listed below) that were manually segmented and visually inspected for quality assurance. These manual segmentations were only used to quantify performance. The total number of subjects was 38: 30 subjects were used for training and 8 subjects for testing. Although we call our sets training and testing, we emphasize that SAE did not have access to the segmentation images during training, as we are proposing an unsupervised paradigm. We repeated the experiment 5 times with different random subject assignments to the train/test partitioning.

### 3.2. Variants of SAE

We employed two atlases. The first one (Atlas1) was based on a single unpaired segmentation image that we obtained from (Marcus et al., 2007), which was automatically segmented using FreeSurfer (Fischl, 2012). We applied spatial blurring (Gaussian with 3 mm isotropic standard deviation) to the one-hot encoded segmentation image to obtain a probabilistic prior. As a second prior (Atlas2), we used a publicly available probabilistic atlas (Puonti et al., 2016), which was computed based on 20 manually labeled subjects. Both priors and all input MRI scans were affine registered to Talairach space. For both of these priors, we implemented two versions: including and excluding the MRF loss of Eq. (10). Specifically, SAE1 (w/o MRF) uses the spatial prior derived by smoothing the single OASIS segmentation. SAE1 (w/ MRF) adds the MRF term of Eq. (10), where the pairwise potential function is computed based on the neighborhood statistics in the OASIS segmentation image. Finally, SAE2 uses the probabilistic atlas prior (Puonti et al., 2016), instantiated with and without the MRF loss.

### 3.3. Benchmark Methods

As naive baselines, we used the most probable label at each voxel in the two priors. **Baseline1** corresponds to Atlas1 and **Baseline2** corresponds to Atlas2. As a strong baseline, we used an implementation of a widely-used atlas-based brain MRI segmentation tool (Van Leemput et al., 1999), which uses Expectation-Maximization (EM) (Dempster et al., 1977) to invert a probabilistic generative model. This EM baseline was run with the two atlases, which we refer to as **EM1** and **EM2**. For each image, the EM baseline numerically solves an optimization problem and is thus relatively slow. Finally, all the data in the EM baseline has been pre-processed the exact way as we did for our model.

As an effective upper bound on performance, we also implemented a supervised model, where a 3D U-Net (Ronneberger et al., 2015) with the same settings as our encoder was trained with the paired manual segmentations in the training data. Negative generalized (soft) Dice (Sudre et al., 2017) was used as the loss function and 6 of the 30 training subjects were reserved for validation. Training was terminated when validation loss stopped improving. As with our previous setup, we repeated this experiment 5 times with different train (N=24), validation (N=6), and test (N=8) splits[1].

### 3.4. Metrics

All presented results are computed on the test images of each round. For quantitative evaluation, we rely on two metrics: the Dice score that measures the volumetric overlap between the automatic segmentation and the ground truth manual segmentation; and the 95%-Hausdorff distance (HD) that quantifies the distance between the boundaries of the automatic and manual segmentations. When the two segmentation maps are exactly the same, Dice score will achieve its maximum value of 1 and HD will be equal to zero.

---

1. Test subjects are always the same for all methods

|  | **Performance Measure** | |
|---|---|---|
| Model | Haussdorff (mm) | Dice Overlap (%) |
| Baseline1 | 4.11±0.07 | 62.82±0.53 |
| EM1 Baseline | 4.25±0.09 | 71.24±0.71 |
| Baseline2 | 3.50±0.06 | 71.45±0.65 |
| SAE1 (w/o MRF) | 3.88±0.05 | 74.64±0.30 |
| SAE1 (w MRF) | 3.81±0.05 | 75.36±0.32 |
| EM2 Baseline | 2.65±0.05 | 79.70±0.54 |
| SAE2 (w/o MRF) | 2.73±0.04 | 79.94±0.34 |
| SAE2 (w MRF) | 2.68±0.05 | 80.54±0.36 |
| Supervised | 2.23±0.07 | 84.60±0.26 |

| Model | Test Time (s) |
|---|---|
| EM | 61.07 |
| SAE (CPU) | 6.58 |
| SAE (GPU) | 1.58 |

Table 1: Mean performance of all methods with their standard errors and computational time per volume at testing.

### 3.5. Experimental Results

Table 1 lists the global average Dice and HD values for the baselines and SAE variants. Regional and subject-level results are also presented in Fig.2. We observe that in every single case and region, SAE produces segmentations that are better than the naive baselines. SAE Dice scores, overall, were 8-12 points higher than the naive atlas based baselines and slightly better than the strong EM baselines. On a modern CPU, the EM baseline had a run-time of around 60 seconds, whereas SAE took less than 7 seconds per single volume at test time (less than 2 sec on a GPU). This represents more than a 10x speed-up over a popular brain MRI segmentation tool, with no discernible reduction in the quality of results.

For SAE, we observe that the adopted prior has a significant impact on the results. With a superior prior, SAE2 (derived from multiple subjects) yields substantially better results than SAE1. In addition, adding spatial consistency via the MRF loss improves the accuracy in all model variants (paired t-test $p < 1e - 6$, for both atlases). This result highlights the importance of having a sophisticated prior. The best unsupervised model, SAE2 (w/ MRF), yielded a Dice score that was about 4 points below the fully supervised model, which is a strong upper bound in our experiment.

A qualitative visualization of SAE2 (w MRF) results is provided in Fig.3. We can see that despite having a fixed prior $p(\mathbf{s})$, our model is able to capture inter-subject neuroanatomical variation. This is mainly due to the decoder, which enforces the latent representation to be useful for reconstruction.

### 4. Discussion

We introduced SAE, a flexible deep learning framework that can be used to train image segmentation models with minimal supervision. We applied SAE to segment brain MRI scans, relying on an unpaired atlas prior. Importantly, SAE does not need manual segmentations paired with the images, which opens up to possibility to deploy it on new imaging techniques, e.g., with high resolution or different contrast. Empirically, we presented the change in segmentation accuracy as we use different types of priors.

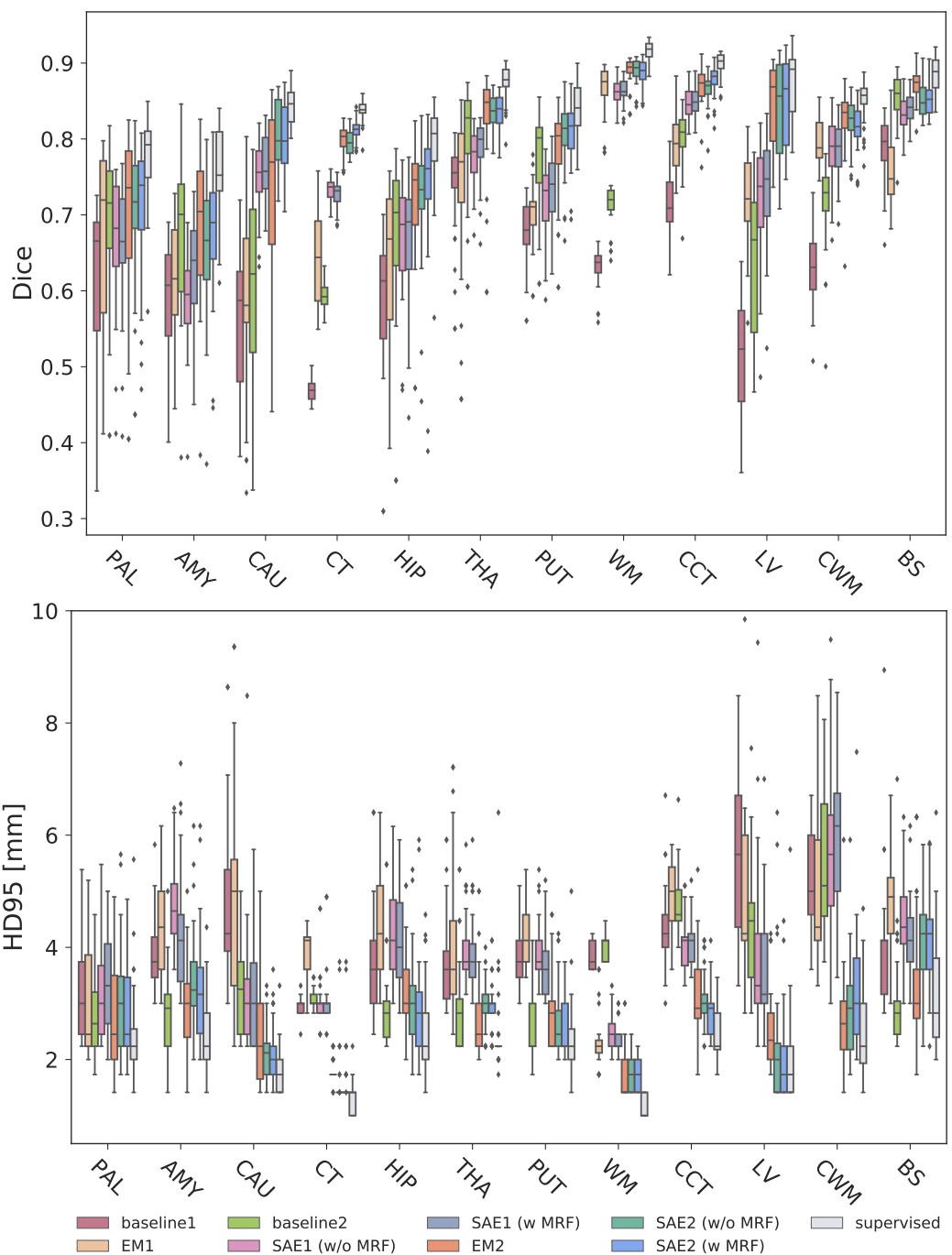

Figure 2: Boxplot of dice and Hausdorff distance. Legend: (PAL) pallidum (AMY) amygdala (CAU) caudate (CT) cerebral cortex (HIP) hippocampus (THA) thalamus (PUT) putamen (WM) white matter (CCT) cerebellar cortex (LV) left ventricle (CMW) cerebral white matter (BS) brainstem.

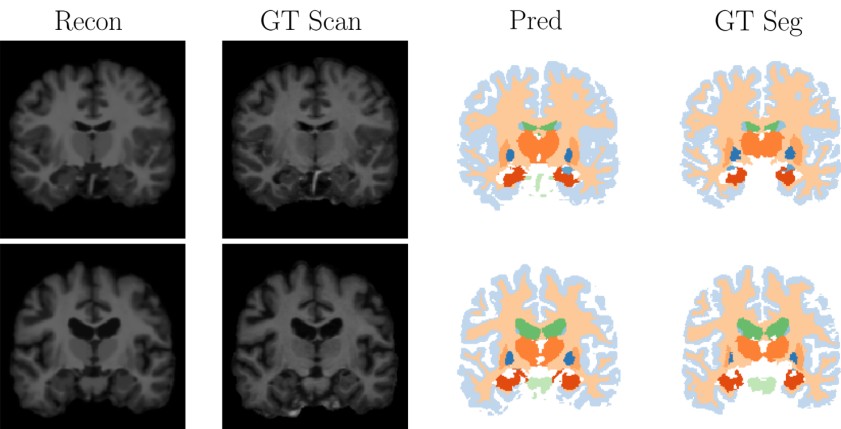

Recon     GT Scan     Pred     GT Seg

Figure 3: Representative segmentation results obtained with SAE2 (w/ MRF) on two subjects. Recon is the output of the decoder. GT scan and segmentation are the input MRI and manual segmentation, respectively. Pred is the segmentation obtained through $\mathtt{argmax}$ of the one-hot encoding $q_\phi(s|x^{(i)})$.

Current implementation of SAE assumes that the input MRI is affine normalized with the prior by working in Talairach space. However, SAE can be implemented with very different types of priors, which we would like to explore in the future. For example, in the present paper, we did not experiment with a spatial deformation model that would warp the atlas to better align with the input image. We envision that we can integrate a "spatial transformer" type neural networks, such as VoxelMorph (Dalca et al., 2018a), to relax our assumption. By adding a deformation model to the prior, we believe that we can handle complications like moving organs. Alternatively, we can implement more sophisticated priors, such as those that exploit an adversarial strategy, as in adversarial autoencoders (Makhzani et al., 2015).

## 5. Acknowledgement

This research was funded by NIH grants 1R21AG050122, R01LM012719, R01AG053949; and, NSF CAREER 1748377, and NSF NeuroNex Grant1707312. JEI is supported by the European Research Council (ERC Starting Grant 677697, project BUNGEE-TOOLS). AVD is supported by NIH 1R56AG064027.

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
