# OpenReview forum: "An Auto-Encoder Strategy for Adaptive Image Segmentation"
_MIDL.io/2020/Conference — MIDL 2020_

### Official Review · AnonReviewer3 · 2020-02-26
**A variational autoencoder segmentation stratege. Novelty is limited, but the result seems good and interesting.**

**Rating:** 3
**Confidence:** 5
**Recommendation:** Poster

**Summary:**

The goal of this work is to alleviate the annotation work of the training data for supervised learning, and make use of the existed segmentation atlas.
The authors proposed a variational autoencoder segmentation strategy. It takes the segmentation as the latent feature, and atlas as the feature prior. Using the idea of VAE, the output of the encoder was forced to be close to the prior, and can be decoded into images as similar as the original input of the encoder. Thus the atlas or the prior is of great effect in practice.
The method was evaluated on brain MRI scans, and compared to EM, it achieved promising results.


**Strengths:**

There are mainly two strengths:
1.	The strategy proposed in this paper can be seen as an atlas-based method. As using networks to predict the segmentation, it takes less time in test stage.
2.	The method can make use of the prior knowledge. As shown in this paper, both the probability map and the MRF knowledge were taken into account, and achieved improvement in segmentation accuracy.


**Weaknesses:**

1 As described in Section 3.1, all data were preprocessed before training or testing, including affine registration, which also took some time. Hence, the computation time can be more in practice, as discussed in Section 3.5.
2 The prior, such as the probability map, has too great effects on the results. In theory, the prior in training stage should be the prior of the test data, hence the atlas should be selected in the same distribution of the test data, which can be different from the Atlas1 or Atlas2 in Section 3.2.
3.  When the prior p(l) is known, the more direct loss would be the MSE loss of the expectation of the output probability and the prior p(l), i.e.,
||E(q(l│x)-p(l)||
This can be taken as the deep learning-based baseline, compared with the variational version.


**Detailed Comments:**

Typos:
1 s_ij in Eq.(5) should be s_ik
2 l_y in Eq. (10) should be l_k


**Justification Of Rating:**

This work developed a deep learning-based strategy to utilize the prior knowledge for image segmentation. They use the idea of VAE, and force the segmentation to be close to the prior, and further can be decoded into images similar to the original one. Hence the prior is of great effect. In experiments, they use the atlas probability map and the MRF to deduce the prior distribution, and achieved improvements in accuracy.
The method is an atlas-based approach with neural network, and given a framework to take the prior knowledge into account. The method is quite interesting, and the paper is well written and easy understanding. However, the novelty of this work is limited.

**Paper Type:**

methodological development

**Questions To Address In The Rebuttal:**

1 report the computational time of the preprocessing of the test data;
2 report the results on test data which had not been preprocessed. This is related to the robustness of the proposed method.
3 report the results of the baseline method described in the third Weakness point.


**Special Issue:**

no

---

> ### Author Response · Authors · 2020-03-26
> **Response to Reviewer 3**
>
> Thank you for taking the time to write a review and provide insightful comments. We would be happy to add any missing relevant citation. A detail response to your questions is provided below:
>
> R3.A1: This is correct. The preprocessing time can take a few minutes per MRI volume. Our model and all baselines relied on the same preprocessing and thus have a fixed upfront cost, which can likely be reduced with more efficient preprocessing tools. Our takeaway message in section 3.5 is that our model is substantially faster than the EM baseline. In the future, we also plan to deploy our model in data that have not been preprocessed.
>
> R3:A2: We agree that the prior plays a critical role in the performance of our proposed method. In fact, the affine pre-registration assumption is crucial in the success of the adopted prior. Without this step, we anticipate that test performance will be unacceptably low. We will emphasize this assumption of our model. However, we also note that the pre-processing steps we used are relatively straightforward in brain MRI processing.
>
> We are currently exploring the use of flexible priors that, for example, have a built-in deformation model, as described in [1]. By adding a spatial transformer (deformation model) to the prior, we believe that we can handle misalignments between the prior and the image. Alternatively, one could use a prior that does not depend on spatial alignment (as in the adversarial learning approach, such as the one used in [2]). We leave these explorations to future research. We will comment on these issues in the updated paper.
>
> [1] Balakrishnan, Guha, et al. "VoxelMorph: a learning framework for deformable medical image registration." IEEE transactions on medical imaging 38.8 (2019): 1788-1800.
> [2] Mescheder et al. "Adversarial variational bayes: Unifying variational autoencoders and generative adversarial networks." 2017
>
> R3.A3: We are not sure we understand this comment. Is the reviewer referring to the situation where the segmentation of an image is known (during training)? If so, this is similar to our supervised model baseline. During our experiment with the supervised baseline, we tried using MSE loss, cross-entropy, and Dice. The supervised model that gave the best results used the Dice loss. Therefore, those are the ones we presented. We are happy to address this point if this comment is expanded.
>
> R3.A4: Thanks for pointing that typo out. It will be corrected in the next revision.

---

### Official Review · AnonReviewer2 · 2020-03-14
**Well-explained approach to unsupervised image segmentation**

**Rating:** 4
**Confidence:** 4
**Recommendation:** Oral, Poster

**Summary:**

This paper proposes use an auto-encoder to produce segmentations, guided by an atlas prior.
The encoder represents the segmentation network, while the decoder reconstructs an image from a segmentation.
Two priors are experimented upon, a pixel-independent class-prior, and a pair-wise MRF prior.
This is implemented by the use of a KL with a latent distribution q, for computational tractability.
The framework allows the use of unpaired images and segmentations, which is useful in practice.

**Strengths:**

- The paper is very clear, well organised, and very well written. Ideas are easy to follow
- Variants are proposed, to show the flexibility of the SAE: 2 different priors, and two different ways of obtaining the Atlas (derived from multiple subjects or just one)
- The experiments are well validated, compared to pertinent methods, and using multiple iterations (5).
- From a methodological standpoint, SAE does not need paired segmentations / images, and is flexible to many priors, which looks promising for future work.

**Weaknesses:**

The paper presents no major weaknesses.
In Figure 2, it is a little difficult to see which boxplots refer to a proposed method, and which refer to an upper/lower baseline or a benchark method. The variablity of regional results ((PAL, AMY,CAU, CT, HIP, ..) could have been discussed a little more.

**Justification Of Rating:**

The paper proposes an principled and flexible framework for which has the interesting benefit of relieving from the necessity of paired images/manual segmentations. It is well organised, well written, and easy to follow.

**Paper Type:**

methodological development

**Special Issue:**

yes

---

> ### Author Response · Authors · 2020-03-26
> **Response to Reviewer 2**
>
> R2.1: Thank you for taking your time to write a review and your positive comments. We agree that it can be difficult to read the boxplots. We will work on making that figure more informative/visible, namely choosing a more contrastive color palette and larger fonts. We will also add some text to discuss the variability across regions. Each brain region tends to follow the trend shown in Table 1. In general, a good prior gives better results. In addition, adding an MRF to the prior, tends to increase the accuracy for each region.

---

### Official Review · AnonReviewer1 · 2020-03-16

**Rating:** 3
**Confidence:** 4

**Summary:**

This paper proposes an original variational auto-encoder framework for segmenting images based solely on an atlas prior. The method is indeed completely unsupervised, no segmentation labels are needed for the training images. The paper is clear and the mathematical method well explained (even if more implementation details are needed). Results show the effectiveness of the proposed method but they are not completely convincing and some details need to be clarified.

**Strengths:**

- Authors present an interesting mathematical model to integrate an atlas prior into a variational auto-encoder segmentation network.
- The paper is well-written, well-organised and easy to read.
- The method is tested on brain images but it could be extended to other organs.

**Weaknesses:**

- As in classical atlas-based segmentation methods, the registration is a key-point. If the atlas is not correctly registered to the test images, the segmentation can not be accurate.  Even if authors mention this pont in the conclusion as future work, this should have been better discussed in the paper. For instance, the baseline EM method, does it use the same registration technique as pre-processing ?
- One of the main results of this paper is the shorter computational time with respect to the EM baseline method. Indeed, the accuracies are quite similar. I think that it would be important to add computational times in Table 1 to make this point clearer to the reader.
- Some implementation details are not well explained. For instance, authors should briefly explain the Gumbel-softmax relaxation scheme. Furthermore, how is \sigma^2 initialised ? Especially for the first 16 subjects ? Why not using an inverse-Wishart prior for \sigma^2 ?
- In the discussion, authors mention that the proposed method "opens up to possibility to deploy it on new imaging techniques". How exactly ? If the atlas prior is not of the same imaging technique as the test or training images, how would that be possible ? Please comment on that.

**Justification Of Rating:**

This paper presents a new and interesting method for using an atlas prior in deep learning segmentation. However, some points are not clearly addressed (see Weaknesses). Most importantly, computational times should be added to Table 1 to make results more convincing.

**Paper Type:**

methodological development

**Special Issue:**

no

---

> ### Author Response · Authors · 2020-03-26
> **Response to Reviewer 1**
>
> Thank you for taking the time to write a review and provide insightful comments. A detail response to your questions is provided below:
>
> R1.A1: The EM baseline has been pre-processed the exact way as we did for our model.
>
> The framework proposed in our paper assumes that the input image is roughly aligned with the prior (at least with a global transformation that brings structure boundaries into close proximity). We are currently exploring the use of flexible priors that, for example, have a built-in deformation model, as described in [1,2]. By adding a spatial transformer (deformation model) to the prior, we believe that we can handle misalignments between the prior and the image. Alternatively, one could use a prior that does not depend on spatial alignment (for example, an adversarial prior, such as the one used in [3]). We leave these explorations to future research. We will comment on these issues in the updated paper.
>
> [1] Balakrishnan, Guha, et al. "VoxelMorph: a learning framework for deformable medical image registration." IEEE transactions on medical imaging 38.8 (2019): 1788-1800.
> [2] Dalca et al. "Unsupervised deep learning for Bayesian brain MRI segmentation." MICCAI 2019
> [3] Mescheder et al. "Adversarial variational bayes: Unifying variational autoencoders and generative adversarial networks." 2017
>
> R1.A2: Our model is indeed faster than EM basilines method during testing. We will recompute and add exact run times in a new table next to Table1. In this new table, the average inference time for SAE on a CPU is 6.58s, SAE on a GPU is 1.58s and the EM algorithm is 61.07s. Hence the SAE model offers a >10x speed-up.
>
> R1.A3: We apologize for the missing details. We will add a brief discussion of the Gumbel-softmax, namely:
>
> The Gumbel-softmax approximates the non-differentiable argmax (tresholding) operation with a softmax in order to make the function differentiable. It provides us with a simple and efficient way to perform the reparameterization trick for a categorical distribution, allowing the network to be trained via back-propagation.
>
> For \sigma^2 initialization, for the first 16 subjects, we set the weight on the reconstruction loss to be zero (effectively setting \sigma^2 to infinity) so that the segmentation (encoder) network was trained only based on the prior. In subsequent batches, \sigma^2 is updated with the running variance, which at the beginning is very high (due to poor reconstructions). However, as reconstructions improve, the variance decreases.
>
> We will clarify these points in the paper.
>
> R1.A4: For training, our proposed method only requires a label atlas prior (and no paired image such as an MRI scan), which can be derived from a single segmentation volume. Note that In our definition, we refer to an atlas as a volume of label probabilities. In fact, the label atlas prior can be created using any technique. This is very different from methods that rely on supervision. Those methods require *paired images and labels (segmentions)* for training. In contrast, unsupervised models, such as the one we present here, can be deployed in a way where models can be continuously updated, without need for human input.

---

### Official Review · AnonReviewer4 · 2020-03-16
**Review of "An Auto-Encoder Strategy for Adaptive Image Segmentation"**

**Rating:** 3
**Confidence:** 4
**Recommendation:** Poster

**Summary:**

This paper uses a variational auto encoder (a U-net) combined with a decoder (a CNN) to address the problem of image segmentation with only weak supervision. The context here is applicability to unlabeled medical image volumes. The notion of "weak" here is the assumption of an available prior on segmentation labels, with an otherwise unlabeled training dataset. The method is described in Section 2 and is tested with a naive prior and an MRF prior, on a dataset of 38 3D MRI scans. The main empirical finding is that this form of weak supervision improves performance over a naive baseline and that the MRF prior performs better than the weak one.

**Strengths:**

The main strength is that the method is described clearly enough in Section 2, and that it appears to be an almost direct application of VAEs (Kingma and Welling 2013), with associated design choices. Thus, though methodological novelty is quite modest, the approach seems well motivated. The paper is well written for the most part, with a few minor typos here and there (probable, an extra or missing article here and there, etc.). The results demonstrate a type of proof of concept. The paper could be improved.

**Weaknesses:**

I'm a bit concerned that the ideas promoted here are rather standard by now, in that VAEs are used for all sorts of problem domains where unlabeled data is not available, and where the basic idea is to evaluate an encoding (in this case a segmentation) based on the error of a decoding (in this case a reconstruction), using the prior. The particular application here illustrates the proof of concept, but there are many assumptions that are not made clear at the outset. What is done is clear enough, but why is not always obvious, and the underlying limitations/assumptions are not adequately discussed. A key issue here is the need for an improved discussion of the notions of a "good" prior.

**Detailed Comments:**

It seems to be that this paper could be strengthened with an editorial revision that more clearly states what the methodological innovation is, and with stronger empirical work. I was not terribly surprised at the improved performance over the naive baselines, since clearly some context should be learned via the encoder-decoder strategy. But I am puzzled that the authors did not go after generalizability as well, i.e., why not throw in data sets where image shifts have occurred to show case robustness in such settings, as the abstract gave a hint of. It is promising though that the weakly supervised method is slowly approaching the performance of a supervised one, at least as far as I can tell from the results in Table 1.

**Justification Of Rating:**

The results, though proof of concept are sound, and I think the authors are onto something interesting. I do worry though that this is the type of thing that many in the ML applied to medical imaging community are doing. I worry too that the basic assumptions were not clearly stated up front. This does not appear to a general method, but rather, one that could give plausible results when the assumptions are met. Many decision choices are not fully motivated. Finally, the choice of suitable priors is itself an interesting problem to nail down.

**Paper Type:**

both

**Questions To Address In The Rebuttal:**

1) In your abstract you claim that methods trained with heavy supervision are sensitive to shifts in image characteristics, e.g., in the context of segmentation the use of different scanners. But as far as I could tell, in your experiments you did not really come back to this point to show that segmentation with weak supervision using a VAE is robust in this setting.

2) Can you clarify what your methodological innovation is? It seems to me a rather direct application of VAEs. It was unclear to me what is new here. It seems to me that the ideas in Fig. 1 are used in a LOT of projects, not necessarily leading to the need for a new paper to be written!

3) Can you discuss how you would learn suitable priors in general for this strategy? You talk about a spatially independent one (which doesn't make a lot of sense in the case of anatomical structure segmentation) and then you use an MRF based prior, which is based on locality, and where the basic idea is not new. Could you optimize the learning of a prior for this formulation? Much in your paper rests on the choice of what you refer to as a "good" prior.

4) Why is additive Gaussian noise (beyond being convenient for the formulation) a valid model for p_\theta(x/s)? Can you discuss this further?

5) Your discussion of a spatial prior only makes sense for tasks where registered data exists. It would be very sensitive to deformations, moving organs, etc. In fact, the entire framework, as far as I can tell, would fail for deformable structure segmentation (or would it?). It would help to more clearly state your assumptions about the particular class of segmentation problems you are proposing to handle at the outset.

6) Why didn't you scale your experiments to apply to larger databases? There are many such datasets, e.g, those used in the MICCAI community for a variety of challenges. You could use them without labels, to strengthen our claims.  Why didn't you also demonstrate robustness to image shifts, degradations, scanner variability, and so on, which is one of the motivating points in the abstract?

7) Why do you not discuss Figures 1, 2, and 3 at all in the main text? (Or if you did I apologize, but I missed this.)

**Special Issue:**

no

---

> ### Author Response · Authors · 2020-03-26
> **Response to Reviewer 4 part 1**
>
> Thank you for taking the time to write a review and provide insightful comments. We would be happy to add any missing relevant citation. In general, we agree that the core design choice in our framework is the prior and we are keen to make that point clear in the paper. A detail response to your questions is provided below.
>
> R4.A1: In the abstract, we did not mean to set a direct comparison between the proposed VAE strategy and a supervised approach under different scanner or domain conditions; but rather we want to motivate why using a model that merely relies on a label prior (and not paired images and labels) can be very powerful and useful. That said, we agree with the reviewer that the quoted claim in the abstract is not substantiated with experiments and we will remove it in the revised version.
>
> R4.A2: The core novelty of our paper, we believe, is in the way we formulate the biomedical image segmentation problem and assemble state-of-the-art deep learning ideas into an effective computational framework. We emphasize that we do not claim any methodological innovation in any of the individual parts that make up our machine learning strategy. That said there are some important differences compared to prior work that we highlight below.
>
> Most popular VAE-based methods, even in the context of biomedical image computing, use a low-dimensional continuous bottleneck representation that follows a Gaussian prior [10, 11, 12]. Original VAE methods were mostly applied to compression, synthesis, or denoising problems. Recently, there has been a lot of on-going work to develop techniques to increase the flexibility of the posterior, the interpretability of the representation, or the quality of the synthesized data. For example, researchers have used auxiliary latent variables [1,2,3] , normalizing flows [4,5], GANS [6] and discrete latent representations [7,8,9].
>
> Our proposed model is different to the aforementioned methods. We are not using an arbitrary low dimensional representation. In the segmentation task, the representation is our ultimate objective and, due to its nature, needs to be treated as a discrete variable that lives on the same pixel grid as the input image. In this manner, we have an interpretable latent space and a domain-specific prior, and we are not interested in compression. Moreover to implement back-propagation through the segmentation layer, we employ the Gumbel softmax/Concrete distribution relaxation.
>
> [1] Maaløe, Lars, et al. "Auxiliary deep generative models." 2016
> [2] Salimans et al. "Markov chain monte carlo and variational inference: Bridging the gap." 2015
> [3] Ranganath et al. "Hierarchical variational models." 2016
> [4] Rezende and Shakir. "Variational inference with normalizing flows." 2015
> [5] Kingma, Durk P., et al. "Improved variational inference with inverse autoregressive flow." 2016
> [6] Mescheder et al. "Adversarial variational bayes: Unifying variational autoencoders and generative adversarial networks." 2017
> [7] Maddison et al. "The concrete distribution: A continuous relaxation of discrete random variables." 2016
> [8] Jang et al. "Categorical reparameterization with gumbel-softmax." 2016
> [9] Razavi et al. "Generating diverse high-fidelity images with vq-vae-2." 2019
> [10] Kohl, Simon, et al. "A probabilistic u-net for segmentation of ambiguous images." 2018
> [11] Myronenko, Andriy. "3D MRI brain tumor segmentation using autoencoder regularization." 2018
> [12] Han, Kuan, et al. "Variational autoencoder: An unsupervised model for modeling and decoding fMRI activity in visual cortex." 2018
>
> R4.A3: We agree that the prior plays a critical role in our formulation. We are interested in exploring alternative approaches. That said, the relatively simple priors we experiment with in this paper yield a tractable loss function and promising results.
>
> One extension to our current prior would be to adopt a non-linear deformation model (as in [13]) that aligns the atlas with each individual subject. The deformation network can be trained via back-propagation during training.
>
> Another approach could be to use a collection of unpaired segmentation images and an adversarial learning approach. Here, a classifier would discriminate the automatic segmentations (output of encoder) and real segmentations. One would then train the VAE on the discrimination loss, as well as the reconstruction loss, similar to [6].
>
> There is also the opportunity to use pre-trained, yet more sophisticated priors than we did in this paper. For example, one could train a separate unsupervised neural network model on unpaired segmentation images, which would provide an estimate of the prior probability.
>
> [13] Dalca, Adrian V., et al. "Unsupervised learning for fast probabilistic diffeomorphic registration." International Conference on Medical Image Computing and Computer-Assisted Intervention. Springer, Cham, 2018.

---

> > ### Author Response · Authors · 2020-03-26
> > **Response to Reviewer 4 part 2**
> >
> > R4.A4: Additive Gaussian noise models are widely used in brain MRI segmentation, e.g. [14,15]. This model assumes that each segmentation region (label) has its own distinct (mean) value and pixel intensities are distributed as an independent Gaussian around that. Since labels in our application correspond to structures that are largely made up of the same tissue type, this is a realistic assumption that has worked well in practice. Our model can also potentially account for partial voluming [14] in boundary pixels, captured by the convolution in the decoder model. We will add text that will address these issues and make our assumption more explicit.
> >
> > [14] Van Leemput et al. "A unifying framework for partial volume segmentation of brain MR images." 2003
> > [15] Wells et al. "Adaptive segmentation of MRI data." 1996
> >
> > R4.A5: Excellent point. The implementation of the proposed framework in this paper assumes that the input image is roughly rigidly aligned with the prior (at least with a global transformation that brings structure boundaries into close proximity). We are currently exploring the use of flexible priors that have a built-in deformation model, as described in R4.A3. By adding a spatial transformer (deformation model) to the prior, we believe that we can handle complications like moving organs. Alternatively, one could use a prior that does not depend on spatial alignment (as in the adversary-based prior, described in R4.A3). We leave these explorations to future research. We will comment on these issues in the updated paper.
> >
> > R4.A6: Even though our model does not use labels for training, we still needed them to evaluate our model. The Buckner dataset has labels that were manually delineated by experts, providing the gold standard to evaluate against. In our experiment, we wanted to emphasize that even if we have a suboptimal atlas (Atlas1), we are able to obtain good performance. As a reminder, Atlas 1 comes from a randomly picked subject in OASIS and had its label automatically generated from FreeSurfer.
> >
> > As we note in R4.A1, we did not mean to set a direct comparison between our model and a supervised model under different scanner or domain conditions, but rather we want to motivate the use of unsupervised methods. We agree that the suggested experiment will be important in showcasing the proposed approach and we plan to do that in an extended journal version of this paper. Also, as mentioned in R4.A1, we will remove the referenced text from the abstract.
> >
> > R4.A7: Fig. 1 is discussed throughout the Method section and mentioned in section 2.3. It provides a pictorial overview of each component of the loss in Eq.5.  Table 1 is a summary of Fig.2, and we discuss the significance of each model in section 3.5. We briefly discussed and reference Fig.3 at the end of section 3.5.

---

### Meta-Review · Area_Chair1 · 2020-04-07
**MetaReview of Paper103 by AreaChair1**

**Rating:** 3
**Recommendation For Accepted Papers:** Poster

**Metareview:**

There is consensus that the technical novelty is limited, but that the results are interesting as a proof of a concept for unsupervised AE segmentation driven by second-order MRF prior combining atlas-based unary and pairwise terms.
(in my opinion, "unsupervised" might be a better term in this case).

When preparing a final version, you should take comments and criticism of the reviews very seriously (particularly for the most detailed first review). Implicit claim of better robustness to scanner variations should be removed. Relationship and differences with standard VAE should be thoroughly discussed. Limitations of your atlas-based prior should be emphasized, as discussed in many of the reviews. All other comments should also be carefully addressed.

**Paper Type:**

both

**Special Issue:**

no

---

### Decision · Program_Chairs · 2020-04-11

Accept